# Conformational plasticity across phylogenetic clusters of RND multidrug efflux pumps and its impact on substrate specificity

Mariya Lazarova[1], Thomas Eicher[1], Clara Börnsen[2], Hui Zeng[1], Mohd Athar [3], Ui Okada [4], Eiki Yamashita [5], Inga M. Spannaus[1], Max Borgosch[1], Hi-jea Cha[1], Attilio V. Vargiu [3], Satoshi Murakami [4] ✉, Kay Diederichs [6] ✉, Achilleas S. Frangakis[2] ✉ & Klaas M. Pos [1] ✉

Antibiotic efflux plays a key role for the multidrug resistance in Gram-negative bacteria. Multidrug efflux pumps of the resistance nodulation and cell division (RND) superfamily function as part of cell envelope spanning systems and provide resistance to diverse antibiotics. Here, we identify two phylogenetic clusters of RND proteins with conserved binding pocket residues and show that the transfer of a single conserved residue between both clusters affects the resistance phenotype not only due to changes in the physicochemical properties of the binding pocket, but also due to an altered equilibrium between the conformational states of the transport cycle. We demonstrate, using single-particle cryo-electron microscopy, that AcrB and OqxB, which represent both clusters, adopt fundamentally different apo states, implying distinct mechanisms for initial substrate binding. The observed conformational plasticity appears phylogenetically conserved and likely plays a role in the diversification of the resistance phenotype among homologous RND pumps.

Active antibiotic export greatly contributes to both intrinsic and acquired resistance in Gram-negative bacteria. While overexpression of drug efflux pumps is often associated with fitness costs, under antibiotic stress it provides an opportunity window for mechanisms of permanent resistance to evolve[1–3]. Resistance nodulation and cell division (RND) efflux pumps are secondary active antiporters that are ubiquitous across all domains of life. As part of tripartite multidrug efflux systems in Gram-negative bacteria, they span the entire cell envelope and export a broad variety of structurally and chemically unrelated toxic substrates[4,5]. The activity of RND efflux pumps is associated with a multidrug resistance phenotype in all clinically relevant Gram-negative bacteria[1,3,6–8].

Knowledge of the structure and function of RND efflux pumps was initially derived from *E. coli* AcrB, one of the best characterised representatives of this superfamily. AcrB forms a homotrimer in the inner membrane and associates with the pore-forming outer membrane factor TolC through the periplasmic adaptor protein AcrA (Supplementary Fig. 1a). Two large periplasmic loops in AcrB form the

[1]Institute of Biochemistry, Goethe-University Frankfurt, Frankfurt, Germany. [2]Buchmann Institute for Molecular Life Sciences and Institute of Biophysics, Goethe-University Frankfurt, Frankfurt, Germany. [3]Department of Physics, University of Cagliari, Cagliari, Italy. [4]Department of Life Science and Technology, Tokyo Institute of Technology, Yokohama, Japan. [5]Institute for Protein Research, Osaka University, Osaka, Japan. [6]Department of Biology, University of Konstanz, Konstanz, Germany. ✉e-mail: murakami@bio.titech.ac.jp; kay.diederichs@uni-konstanz.de; achilleas.frangakis@biophysik.org; pos@em.uni-frankfurt.de

substrate-binding porter domain (PD) and the funnel domain (FD) (Supplementary Fig. 1b). The full assembly of the tripartite system is necessary for efflux activity, while the PD determines substrate specificity. During drug efflux, AcrB undergoes a functional rotation where each of the three protomers sequentially cycles through the conformational states loose (L), tight (T) and open (O). Substrates can enter the PD through several channels and bind to the access pocket (AP) in the L state and the deep binding pocket (DBP) in the T state (Supplementary Fig. 1c). Structures of the AcrB trimer, with each protomer in either the L, T, or O states (LTO trimer), have been elucidated without bound drugs[9,10]. Additionally, structures of LTO trimers with drugs bound in only the T state[9,11,12] or only the L state[12,13] have been determined. Furthermore, binding of different drugs to both the AP and DBP within one LTO trimer, as well as binding of drugs to the upper regions of the TM1/TM2 and TM7/TM8 groove, exemplifies multisite binding within one LTO trimer[11,13]. The binding hub for most drugs, however, is the DBP, even if the drugs bind to the other sites mentioned in the initial phase of the transport. The groove of the DBP in AcrB is lined by hydrophobic, mostly aromatic, residues (Supplementary Fig. 1d). They form an open pocket in the T state that accommodates a wide range of structurally and physicochemically diverse drugs. The substrate is extruded through an exit channel in the O state by a closure of the binding pockets due to rigid-body movement of the porter subdomains. This movement is facilitated by the binding of protons to the titratable D407 and D408 residues located in the transmembrane domain (TMD). The cycle resets via an O to L transition, where the proton is released from the TMD to the cytoplasm[5,9,10,14,15]. In the O and L states, the rearrangements in the PD lead to the collapse of the DBP and a tight packing of the hydrophobic residues (Supplementary Fig. 1d)[9,10,12]. Comparison of apo- and drug-bound structures of AcrB[9,10,16] revealed that the drug-binding AP and DBP adopt their drug-accepting geometry and volume, except for minor but important side chain reorientations, already in the apo state of the L and T protomers. This suggests a conformational selection mechanism, as the structures of the AP or DBP in the apo or drug-bound state in the L or T states, respectively, do not differ, regardless of whether substrates are bound. The L state is essential for the initial binding of high-molecular-weight drugs to the AP, as demonstrated for macrolides and ansamycins[11,13]. Therefore, the reduction or absence of L-state protomers in the AcrB trimer would decrease the number of AP binding sites for these drugs, leading to a reduction in resistance mediated by the AcrAB-TolC pump.

Recent structural studies of RND multidrug efflux pumps from other Gram-negative bacteria show that they share a common structural architecture and general functional principles with *E. coli* AcrB[17–23]. However, the identification of alternative trimer conformations, particularly the OOO states of AdeB from *A. baumannii*[19,20] and CmeB from *C. jejuni*[18], has posed questions about the conservation of the AcrB transport model in other RND pumps. The varying effects of substitutions at homologous residues in the DBP of AdeB and AcrB are also implicated in the observed differences in substrate specificity and resistance profiles between these two transporters[19,24]. AdeB displays a much higher preference for polyaromatic compounds than AcrB and single-site substitutions changing the polyaromatic interaction sites within the DBP greatly decreased pump activity. The AdeB variants with the DBP substitutions that remove polyaromatic substrate-interacting hydrophobic residues, however, confer better resistance toward more hydrophilic compounds, especially levofloxacin and chloramphenicol, concomitant to the substrate preference of AcrB.

Here, we show that substitutions in the DBP do not only have local effects on the physicochemical properties at and around the substitution site, but can also cause changes in the global conformational landscape of AcrB. Moreover, this global conformational change may influence substrate preference for erythromycin or phenicols, thereby resembling the substrate recognition profile of homologous RND

multidrug efflux pumps like OqxB, which are classified within a separate phylogenetic cluster.

## Results

### Conserved DBP substitution alters AcrB resistance phenotype

Gram-negative RND efflux pumps are known for their substrate promiscuity, meaning they can transport a wide range of substrates. However, substrate specificity can vary depending on the individual efflux pump. For instance, *E. coli* AcrB lacks aminoglycoside efflux, while *E. coli* AcrD does not confer resistance to ciprofloxacin, chloramphenicol, or tetracycline[25]. Since the DBP plays a crucial role in determining the drug specificity of these efflux pumps[26], we investigated the conservation of the amino acid residues that form the DBP by a sequence analysis of over 50 RND representatives from Gram-negative pathogens (Supplementary Table 1). Based on the similarity of their full-length sequences, five phylogenetic clusters were identified (Fig. 1a and Supplementary Fig. 2a, b). These clusters were designated as AcrB, OqxB, MexW, VexF, and MdtB clusters, based on a representative RND efflux pump member within each cluster. Notably, only the AcrB and OqxB clusters exhibited highly conserved residues defining the DBP, with the exception of I277 and I626 (Fig. 1b and Supplementary Fig. 2c, d). The AcrB cluster includes AcrB and the closely related AcrB homologue MdtF from *E. coli*, while the OqxB cluster includes OqxB from *Klebsiella pneumoniae* and BpeF from *Burkholderia pseudomallei*, among others (Supplementary Fig. 2a, b). We observed that despite the conservation within the DBP of AcrB and OqxB cluster members, positions F610 and V612 in the members of the AcrB cluster are exchanged in the members of the OqxB cluster (Fig. 1b and Supplementary Fig. 2c, d).

The V612F exchange caught our attention as a previous evolutionary study[27] demonstrated that under antibiotic pressure, MdtF from the AcrB cluster naturally acquires this substitution. This results in an increased resistance to linezolid, tetracycline, chloramphenicol, and fluoroquinolones, but a reduced resistance to macrolides[27]. Interestingly, the resistance profile of this MdtF variant mirrors that of the OqxB cluster representatives, i.e. OqxB, BpeF, AdeG, and MexF, which confer resistance to tetracyclines, chloramphenicol, and fluoroquinolones, but not macrolides (Supplementary Fig. 3)[21,22,28–30]. Previous studies have shown that drug resistance profiles are generally insensitive to DBP side chain substitutions[19,31]. However, for certain drugs, the DBP side chains F178, F615, and F617 have been found to result in lower MICs for cells harbouring these variants. As an exception, a substitution of F610 with A has major impact on the resistance of most tested drug substrates, including rhodamine 6G, TPP, levofloxacin, tetracycline, erythromycin and chloramphenicol, amongst others[19,31]. Molecular dynamics (MD) simulations[32] offered a mechanistic explanation for this phenotype and suggested that F610 in the wildtype DBP prevents a drug such as doxorubicin to slide deeply into the hydrophobic trap within the DBP. In the F610A variant, such a sliding tightens the binding, and it might hinder the substrate release during the T to O transition. Our attention was therefore focused on the AcrB V612.

Since the substitution of V to F in the MdtF protein from the AcrB cluster resulted in a substrate specificity profile similar to that of the members of the OqxB cluster[27], we sought to confirm this phenotype by substituting the V612 residue of AcrB from *E. coli*, as MdtF and AcrB belong to the same cluster.

We substituted V612 in AcrB with F to mimic the sequence in the OqxB cluster and with a physicochemical similar (W) and different (N, A) residues. We tested the resistance phenotypes of wildtype AcrB and the V612 variants against a panel of 20 toxic substrates (Fig. 1c, Supplementary Fig. 4, Supplementary Tables 2 and 3). The presence of AcrB wildtype and the variants in the membrane was detected via Western blot analysis (Supplementary Fig. 4a). Susceptibility against the panel of drugs was either determined via plate dilution assays[19]

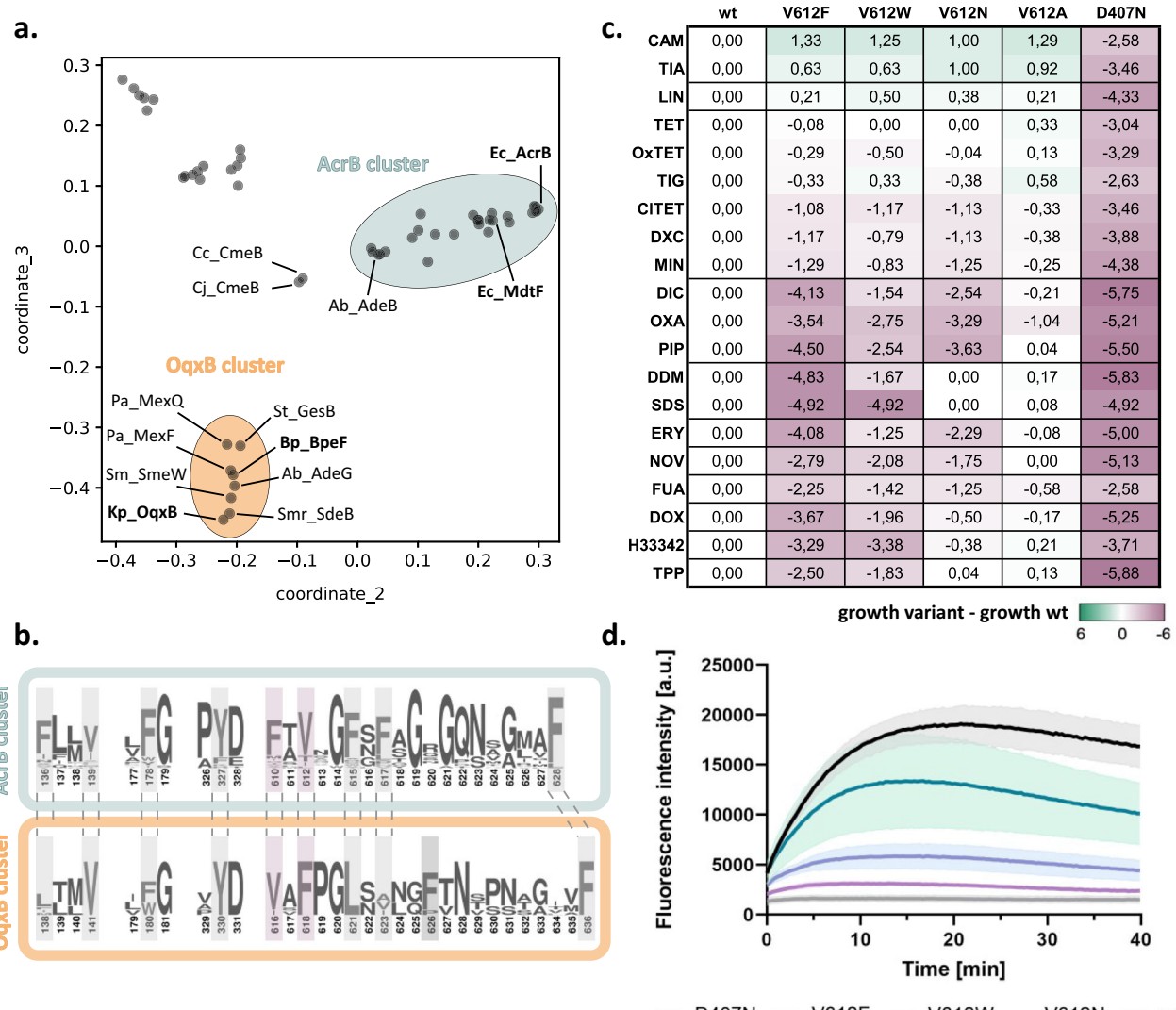

**Fig. 1 | A conserved DBP residue alters the resistance phenotype conferred by _E. coli_ AcrB. a** Map of pairwise sequence similarities (PaSiMap[48]) between representative RND proteins (Supplementary Table 1). The coordinates for the two highest dimensions (coordinate_2 and coordinate_3) are displayed in the plot. The AcrB and OqxB clusters are highlighted in cyan and orange, respectively. Abbreviations are given in Supplementary Fig. 2. **b** Consensus sequence of the AcrB (cyan) and OqxB (orange) clusters. Residues that are part of the DBP are highlighted in grey with AcrB_F610 and V612 in purple. Residue numbers correspond to the sequence of AcrB or OqxB, respectively. **c** Phenotype characterisation of AcrB V612 variants by plate dilution assays. The last dilution step for which growth was

detected was determined and normalised to the wildtype (wt). The inactive D407N was used as a negative control. Green: increased growth, purple: decreased growth; abbreviation as in Supplementary Table 2. The figure shows average data of three biological replicates. **d** Berberine accumulation in _E. coli_ cells expressing different AcrB V612 variants. AcrB activity was monitored by measurement of the berberine fluorescence. AcrB wildtype (wt) and the inactive D407N were used as controls. Data present the mean values (solid line) with standard deviation (shaded background) of three biological replicates. Source data are provided as a Source data file.

(Fig. 1c) or via MIC determination (Supplementary Fig. 4b and Supplementary Table 3).

All V612 variants showed a small but highly reproducible increase in resistance towards phenicols and linezolid, in line with the phenotype of the MdtF variant and the members of the OqxB cluster[21,22,27–30] (Fig. 1c and Supplementary Fig. 4b). However, resistance for most other tested substrates was decreased for the V612F/W variants, with V612F having a more pronounced phenotype. V612N also showed a similar reduced resistance for many of the tested drugs (Fig. 1c).

To directly assess AcrB-mediated efflux, we performed a whole cell drug transport assay with the fluorescent dye berberine (Fig. 1d). Berberine accumulation inside _E. coli_ cells can be monitored by the increase of fluorescence due to its DNA-intercalating properties. AcrAB-TolC effectively exports berberine resulting in a much lower

fluorescence signal compared to efflux-deficient cells. In agreement with the phenotype assays (Fig. 1c) that suggest compromised activity for the V612 variants, a reduction of berberine efflux was observed (Fig. 1d). Of the tested V612 variants, V612F was the most and V612N the least compromised in berberine efflux, compared to cells expressing wildtype AcrB.

In summary, the V612 variants of AcrB exhibited a slight increase in resistance to phenicols and linezolid. However, they displayed a decreased resistance to most other tested substrates, as well as a reduced berberine efflux. Among the variants, V612F was the most compromised, while V612N showed the least compromised resistance.

**Structures of AcrB V612F/W with minocycline in TTT state**

The DBP residues are directly involved in substrate binding, as was shown for AcrB and further RND efflux pumps[9,12,19,20,22]. Thus, the

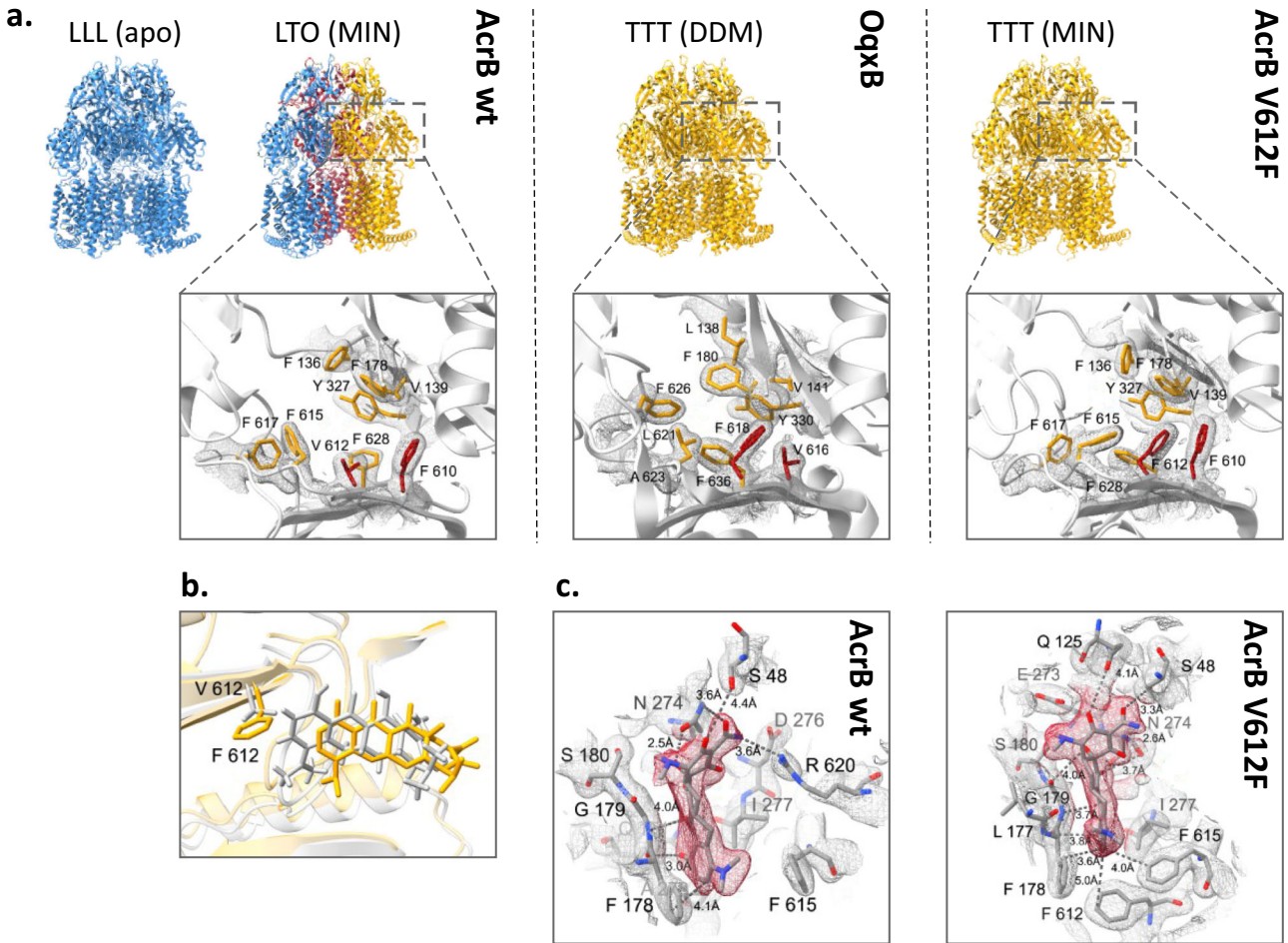

**Fig. 2 | Comparison of the deep binding pocket of AcrB wildtype, AcrB V612F and OqxB. a** Upper panel: Side view of the trimer structures of AcrB wildtype, OqxB, and AcrB_V612F (PDB IDs: AcrB (LLL) – 1iwg. AcrB (LTO) – 4dx5, OqxB – 7cz9, AcrB_V612F – this study). The trimer conformation and DBP-bound substrates are indicated above the trimer. Monomer colours: L state - blue, T state - yellow and O state - red. Abbreviations: MIN minocycline, DDM dodecyl-ß-D-maltoside. Lower panel: top view of the DBP in the T state with conserved DBP residues shown as sticks. Crystallographic $2F_o$-$F_c$ densities are depicted as a mesh contoured at 1 σ.

The residues at positions 610 and 612 in AcrB and the corresponding positions 616 and 618 in OqxB are highlighted in red. **b** Overlay of the minocycline binding pose in the experimental structures of AcrB wildtype (grey, PDB ID: 4dx5) and V612F (this study, yellow). **c** Minocycline interactions in the deep binding pocket of AcrB wildtype (PDB ID: 4dx5) and V612F. The crystallographic $2F_o$-$F_c$ maps are shown at 1 σ (mesh) and the densities for minocycline are highlighted in red. Minocycline and residues with at least one atom within 4 Å distance of the ligand are shown as sticks. Carbon atoms are given in grey, oxygen in red, and nitrogen in blue.

various V612 substitutions alter the substrate binding site, and the observed phenotype change in the AcrB variants may be explained through changed ligand interactions. To assess this, we solved the co-structures of V612F and V612W in complex with minocycline via X-ray crystallography. In contrast to the minocycline structure of AcrB wildtype, that displays an asymmetric LTO with bound minocycline in the T state protomer[9,12], the obtained V612F and V612W co-structures are in the C3 symmetric space group I23 with one AcrB monomer and one DARPin molecule in the asymmetric unit. Thus, the structures represent an AcrB trimer with three identical chains closely fitting the T state of wildtype AcrB (RMSD 1.1 Å for V612F and 1.1 Å for V612W, wildtype reference PDB ID: 4dx5) (Fig. 2a and Supplementary Fig. 5). The introduced F or W side chain is sandwiched between the reoriented F615 and F610, forming a stack of aromatic rings, and closes off the groove of the DBP, thus reducing its size. Minocycline is shifted in the binding pocket (Fig. 2b) to avoid steric overlap with the introduced F or W at position 612. Compared to the wildtype co-structure, the contact between R620 and minocycline is lost, but appears to be compensated by additional H-bonding interactions via carbonyl oxygen acceptors of Q125 and G179 (Fig. 2c and Supplementary Fig. 5c). Further, the flipped F615 is in interaction distance of the aromatic ring

of minocycline. The V612 variant co-structures demonstrate the plasticity of the DBP that is able to accommodate the ligand and allows the formation of alternative interactions despite the alterations in the binding network. Corresponding to these results, the resistance against minocycline is only marginally affected by the V612 substitutions (Fig. 1c).

A computational study was conducted to analyze how further substrates are accommodated in the substituted DBP. The details of this study are available in the Supplementary Notes. The key findings indicate that the V612F and V612W substitutions cause changes in how chloramphenicol and doxorubicin bind to the DBP. These alterations in binding are likely due to modifications in the local structure of the DBP and its interactions with the newly introduced aromatic amino acid residues (F or W at position 612), which are taken into account in docking calculations by enabling flexibility of substrates and of the DBP residues as well as in subsequent MD simulations performed for minocycline and chloramphenicol. This is illustrated in Supplementary Notes for doxorubicin, which can bind to the DBP in the T protomer of both AcrB variants due to a reorientation of F612 (which would otherwise clash with the ligand) and to a minor extent of W612. These changes in binding could explain the observed changes in the

resistance phenotype (Supplementary Fig. 6, Supplementary Table 4, and Supplementary Notes).

The sliding of minocycline and doxorubicin towards the entrance of the DBP indicated by docking in both AcrB variants (Supplementary Fig. 6, Supplementary Notes), suggests a general feature of tetracyclines, highlighting the DBP's plasticity in utilising alternative substrate-protein interaction networks to produce remarkably similar resistance phenotypes, as observed for these compounds (Fig. 1c). These findings are confirmed by multiple MD simulations performed for minocycline, showing that this substrate remains stably bound within the pocket, closely matching experimental pose (RMSD values around 2–3 Å across all MD replicas; see Supplementary Notes). In contrast, the substitutions V612F or V612W significantly shift the smaller, more flexible, and amphipathic chloramphenicol upwards from its binding position in the wildtype transporter (Supplementary Notes). While, in this case, MD simulations reveal pronounced conformational rearrangements of chloramphenicol from the docking pose in both the wildtype and the variants of AcrB (Supplementary Notes), the upward shift of the substrate towards the groove of the DBP is even more pronounced, particularly in the V612F variant (Supplementary Notes). These findings align with previous studies demonstrating that chloramphenicol could accommodate different and possibly resonant subsites of the DBP even in the AcrB wildtype (Supplementary Notes)[32,33]. Moreover, the binding of chloramphenicol in the wildtype AcrB is compatible with experimental findings by cryo-EM[33] (Supplementary Notes). In both variants, the substrate is stabilised by aromatic interactions involving F/W612 (Supplementary Notes, Supplementary Fig. 6 and Supplementary Table 4). We speculate that the slight increase in resistance conferred by V612F and V612W (Fig. 1c) could be attributed to this enhanced stability compared to the wildtype AcrB, leading to optimal retention times triggering functional rotation and enabling more efficient transport. A similar scenario might occur with other flexible, amphipathic, low molecular weight substrates containing an aromatic ring, such as linezolid.

Erythromycin, a high-molecular-weight drug, has been found to bind both at the AP-DBP interface of the L state in *E. coli* AcrB and within the DBP in the T state of its close homologue from *K. pneumoniae* (96% sequence similarity)[11,13,34]. In our top docking pose within the T state of the *E. coli* AcrB wildtype, erythromycin is slightly displaced from the position observed in the *K. pneumoniae* AcrB/erythromycin co-structure[34] as well as the positions of minocycline and doxorubicin (Supplementary Notes), but is closer to this site than to its observed binding site in the AP[13]. The docking pose suggests an intermediate interaction mode along the way from the AP towards the DBP. Since erythromycin is located away from V612, the steric effects due to the substitution are likely less relevant compared to the tetracyclines. Additionally, erythromycin lacks an aromatic ring to potentially interact efficiently with F/W insertions. In agreement with the potential smaller effect of V612F/W substitutions on the interaction network of erythromycin compared to the other investigated substrates, virtually identical binding poses were found for this compound in AcrB wildtype, and the AcrB variants V612F and V612W. Furthermore, binding energies were also similar, with only a minor difference in the variants (Supplementary Notes, Supplementary Fig. 6 and Supplementary Table 4). Since erythromycin is likely sequestered from the periplasm through the AP in the L state[11,13] and subsequently transferred to the PD interior during the L to T state transition, we anticipate that early favourable interactions are reduced if the AP adopts the T state conformation, as proposed earlier[32].

In contrast to wildtype AcrB, that crystallises in the LLL and LTO states[9–13,35], the V612F/W crystal structures were exclusively obtained in the TTT state (Fig. 2 and Supplementary Fig. 5). We thus hypothesised that the substitution impedes the formation of the L state and this in turn might play a role for the transport of L state-binding drugs such as erythromycin. The TTT conformation has previously been shown for AcrB wildtype in a single-particle cryogenic electron microscopy (cryo-EM) structure of the AcrAB-TolC complex with the high affinity inhibitor MBX3132 in the DBP of all three T protomers[36]. As we anticipated that minocycline binding to the DBP might be a driver for the TTT conformation in the V612 variants, we solved the apo structures by X-ray crystallography (Supplementary Fig. 5d). These also adopted the TTT state with an open, but empty DBP. Further, two apo-TTT state crystal structures of representatives from the OqxB cluster, BpeF and OqxB have been described recently[21,22]. The structures of these detergent-solubilized proteins indicated the presence of detergent densities inside the DBP and detergent binding was proposed to induce the observed TTT state[21,22]. Despite the high resolution of our AcrB V612F/W electron density maps (2.3 Å and 2.8 Å, respectively), no clearly assignable detergent (DDM) densities could be observed in the DBP. We therefore assumed that the crystallisation conditions might favour the crystal contacts leading to the TTT state for the AcrB variants. To elucidate the conformation of AcrB without the crystallisation bias, we assessed the structure of the variants by cryo-EM.

## Detergent alters AcrB equilibrium towards the T state

The trimeric states and the distribution of the monomeric conformations of AcrB wildtype and the V612F/W variants were determined via cryo-EM both in a DDM-solubilized and in detergent-free SaliPro nanodisc (SP-ND)[37] reconstituted samples. The processing pipeline and the workflow for the evaluation for each cryo-EM dataset are depicted in Supplementary Figs. 7–15. The processing pipeline involves selecting 2D and 3D classes that exhibit clear structural features and minimal noise, calculating a 3D density map of the trimer, extracting the monomers, and then processing the monomer volumes to determine the distribution of conformational states. For AcrB wildtype solubilised in DDM, an almost even distribution of particles in the L, T and O state was observed with most of the trimers (65.1 %) in the LTO state (Fig. 3a). This is in agreement with the LTO apo-state structures observed by X-ray crystallography[9,10,38]. In contrast, V612F and V612W mainly showed particles in the T state (72.7 % and 83.9 %, respectively), with these variants displaying trimers predominantly in the TTO (54.9 % V612F, 44.1 % V612W) and TTT (31.8 % V612F, 53.8 % V612W) states (Fig. 3a). Notably, no particles in the L state were found for V612F/W.

For wildtype AcrB in SP-ND, a higher abundance of the L state was observed compared to the DDM sample (55.7% in SP-ND versus 37.8% in DDM) (Fig. 3b). Further, the number of trimer particles in the LTO state (43.0 %) decreased, while the abundance of LLO, LLL and LLT states was higher. This suggests an intrinsic flexibility of the AcrB trimer that exists in a dynamic equilibrium between the different conformational states. DDM binding seems to increase the number of T states driving the LTO formation from the LLO, LLL and LLT trimers. Unambiguous DDM densities were not detected in the DBP, however, well-resolved detergent densities were present in the TM1/TM2 groove in the TMD (Supplementary Fig. 16a, b). DDM binding in this groove has been observed in several crystallographic structures of AcrB[11,12] and the TM1/TM2 groove might represent an allosteric binding site or a pocket for initial binding at the entrance of channel 4. In the T state, the PN2 subdomain shifts closer to the membrane plane in comparison to the L state (Supplementary Fig. 16c) and allows interactions of the maltoside headgroup of DDM with the residues N298 and D301. This is specific to the T state since in the L state the PN2 subdomain is in the up conformation and N298 and D301 are not within hydrogen bonding distance of the DDM. Thus, the interactions of DDM in the TM1/TM2 groove might stabilise PN2 architecture of the T state and facilitate the increased formation of T monomers.

For V612F in SP-ND we found that all three states, L, T and O, were present (Fig. 3b), indicating that DDM binding is responsible for the absence of the L state in the detergent-solubilised samples. The T

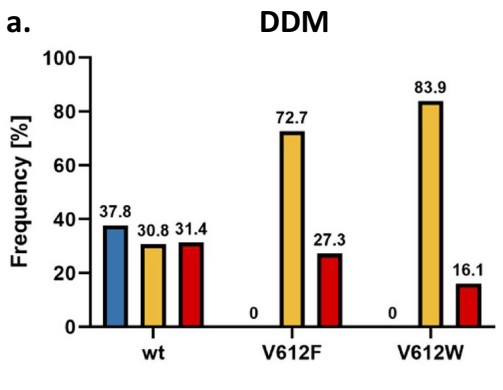

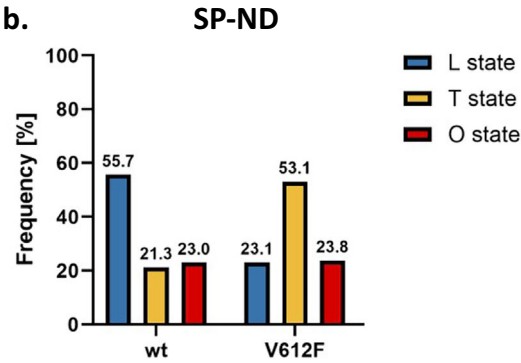

**Fig. 3 | Cryogenic electron microscopy (cryo-EM) analysis of the conformational states of AcrB.** Cryo-EM datasets of AcrB wildtype (wt), V612F and V612W were acquired and the number of particles in the L, T and O conformations was determined as described in Supplementary Fig. 7. The evaluation of each dataset is shown in more detail in Supplementary Figs. 8–13. Summary data for the frequency of each monomer state in the samples of AcrB solubilised in DDM (**a**) and reconstituted in salipro nanodiscs (**b**) are shown in the top panel. The distribution of trimeric states is shown in the bottom panel. The frequency is presented as the percentage of the total number of particles. Source data are provided as a Source data file.

state remains, however, the most abundant state for V612F (53.1% T state in V612F in SP-ND vs 21.3% in the wildtype). The trimer adopts the LTO state, and also the TTO, TTT and TTL states in contrast to the LLO, LLL and LLT states observed for wildtype AcrB. Thus, the introduced substitution clearly shifts the equilibrium between the L and T states in favour of the T state. F/W612 appears to stabilise an open DBP even in the absence of a substrate, as the bulky side chain might mimic the binding of a small substrate. Moreover, the proximity of the bulky aromatic side chains in the hydrophobic cluster might introduce a steric hindrance for the rearrangements associated with the closing of the DBP required for the O and L state formations. Indeed, our structural models of the best resolved O monomer densities show that the DBP remains partially open in V612F/W structure (Supplementary Fig. 17). We assume that the O state conformation is still feasible due to compensating interactions, such as the PC1 and PC2 subdomain proximity, and PN1 subdomain interaction with the neighbouring protomer. However, in the L state, such stabilising contacts are far less pronounced. Thus, the stabilisation of the T state DBP in its open form and impaired DBP closing are likely the reason behind the observed increased abundance of the T state on the expense of the L state in the V612F variant. Detergent binding to the TM1/TM2 groove likely potentiates the shift toward the T state, resulting in the complete absence of the L state in the DDM solubilised samples.

For the V612W variant, a similar structural effect is expected as for V612F due to the introduction of a bulky aromatic side chain in the DBP, corresponding to the TTT crystal structure that was obtained for both V612F and V612W. For the V612N variant we obtained two crystal structures in different space groups, which represent not only the TTT conformation, but also the LTO conformation as seen in wildtype AcrB (Supplementary Figs. 18 and 19). Presumably, here, the closing of the DBP in the L state is also unfavourable due to the introduction of the

hydrophilic asparagine within the aromatic cluster. This is likely less drastic than the effect of the V612F/W substitutions, but could still shift the equilibrium between the L and T states, hence, crystal structures were obtained in both LTO and TTT conformations. The reduction of the abundance of the L state likely affects the transport of substrates that require initial binding in the L protomer, such as erythromycin. Thus, the change of the global conformation of AcrB represents an additional effect of the substitution beyond the direct interactions in the DBP. The observed changes in the phenotype (Fig. 1c, d and Supplementary Fig. 4b) are likely provoked by an interplay of an altered interaction network and a change in the initial binding and entry of the substrate. A detailed analysis would necessitate an in vitro transport system capable of determining quantitative kinetic parameters. Such a system has been described for the homologue *P. aeruginosa* tripartite pump MexAB-OprM[39,40]. While the development of this system has been a groundbreaking assay in the field, it is currently only described for qualitative measurements and with a specific substrate of the pump, ethidium; developing a quantitative transport assay for kinetic analysis of AcrAB-TolC catalysis remains beyond the scope of the current work.

### CryoEM structure of *K. pneumoniae* OqxB
Given the parallels in both the phenotype and the structural characteristics of AcrB V612F and the proteins from the OqxB cluster, we decided to assess the structural characteristics of OqxB as a representative of this cluster. We were able to obtain a detergent-solubilised OqxB crystal structure in the TTO state (Supplementary Table 5 and Supplementary Fig. 20). Thus, OqxB can also adopt an asymmetric structure next to the previously determined TTT state[22] in the presence of a substrate (here: DDM). The O monomer of the TTO structure closely resembles the O state of AcrB and has an open exit channel for

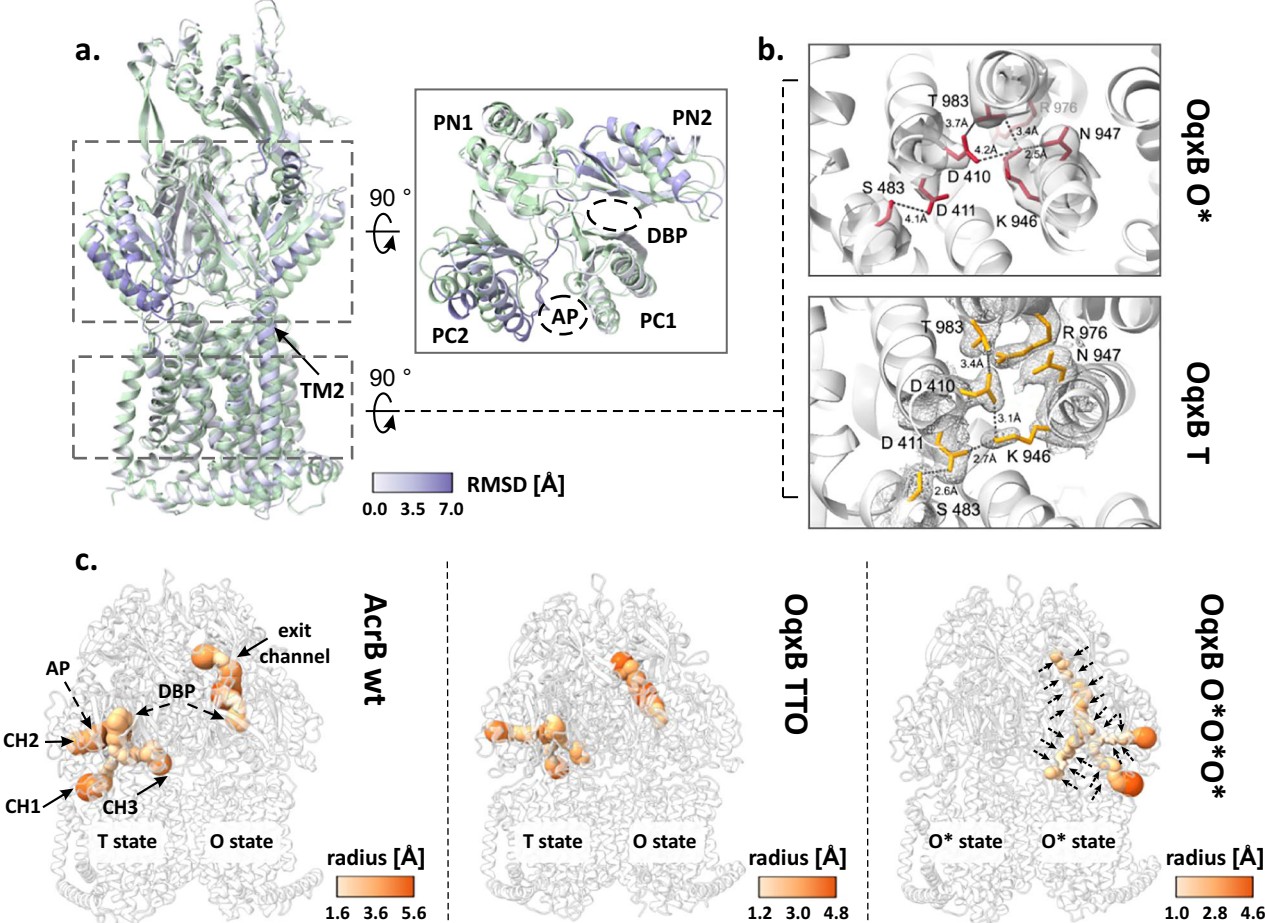

**Fig. 4 | Cryogenic electron microscopy (cryo-EM) structure of OqxB reconstituted in salipro nanodiscs. a, b** Comparison of the OqxB structure in the O* and T states. The O*O*O* cryo-EM structure of OqxB (this study) was overlayed with the previously solved OqxB structure in the TTT state (PDB ID: 7cz9). One monomer of each structure is shown representatively in a. The O* state is coloured by the RSMD between both structures, the T state is coloured green. Right inlet: top view of the porter domain. The transmembrane helix 2 (TM2), the PD subdomains and the access and deep binding pockets (AP and DBP) are highlighted. **b** Proton translocation network in the OqxB O* (top, red) and T (bottom, yellow) states. Crystallographic 2Fo-Fc maps (T state, PDB ID: 7cz9) are depicted at 1σ (mesh). Cryo-EM densities (O* state) are depicted at contour level 0.238 (solid surface). **c** Entry and exit channels in the AcrB and OqxB structures. The channels in the porter domain of the LTO AcrB structure (left panel, PDB ID: 4dx5), the crystallographic structure of OqxB in the TTO state (middle panel, this study) and of the cryo-EM O*O*O* structure of OqxB (right panel, this study) were calculated with MOLE[65]. The channels are shown coloured by radius according to the respective colour key and are labelled representatively in the AcrB structure. The overall decrease in the channels' diameter in the OqxB O*O*O* structure is indicated by the arrows.

substrate extrusion as expected (Fig. 4, Supplementary Table 5). Further, we reconstituted OqxB in SP-ND to assess its distribution of conformational states in a detergent free environment with cryo-EM (Supplementary Fig. 14). In contrast to all AcrB samples, that contained a mixture of several trimeric states, OqxB showed a homogeneous structure with all particle classes representing the same state (Supplementary Fig. 14). Based on the electron density map of the OqxB trimer, a structural model was built with 2.8 Å global resolution (Supplementary Fig. 15). The three individual protomers in the OqxB trimer adopt a highly similar conformation with an all-atom RMSD between the individual chains of ≤ 0.8. In comparison to the T state (reference OqxB_TTT PDB ID: 7cz9), each protomer chain displays an upward shift of the transmembrane helix 2 (TM2) and a tight packing of the subdomains within the PD (Fig. 4a). Further, the central K946 residue of the proton translocation network within the TMD is flipped towards N947 and thus oriented away from both titratable residues D411 and D410 (Fig. 4b). These are characteristics of the O state[5] that were also observed for the O monomer of the crystallographic TTO structure (Supplementary Fig. 20). Therefore, the SP-ND reconstituted OqxB trimer resembles the OOO states observed for AdeB and CmeB[18,19] more closely than the flexible asymmetric AcrB states. However, in all

three monomers of OqxB all channels leading to the PD, including the exit tunnel, are very narrow throughout their entire length with a bottleneck radius between 1.1 Å and 1.5 Å (Fig. 4c). These channels are too narrow to fit any known OqxB substrate. Thus, the cryo-EM structure of OqxB has the typical architecture of the O state, but with a closed exit tunnel and will hereafter be referred to as O*O*O* state. In contrast to the O state observed in the crystallographic TTO structure, the O* state of OqxB has a pronounced shift in the PN1 subdomain (Supplementary Fig. 21), which is likely the reason for the closed configuration of the exit channel.

A monomer state with the characteristic architecture of the O state but with a closed exit channel has been described for several further HAE-1 RND efflux pumps: *B. pseudomallei* BpeB, *C. jejuni* CmeB, *A. baumannii* AdeB and *P. aeruginosa* MexB[18,20,21,23]. A comparison between the different O* states reveals that for BpeB, AdeB and MexB a shift of the PN1 subdomain is observed in the O* state in comparison to the O state similarly to OqxB (Supplementary Fig. 21). This PN1 orientation resembles the conformation of this subdomain in the T state and is likely the reason for the reduced diameter of the exit channel. It has been proposed that the O* state is formed during the transition from O to L[21] and the following model, incorporating the O* state in the

conformational cycle, is feasible: substrates enter the PD through different channels or through the AP and ultimately reach the DBP in the T state. Protonation in the TMD results in the formation of the O state and extrusion of the substrate through the exit channel as previously described[9,10,14]. The presence of the substrate might stabilise the open conformation of the exit channel in the O state. Next, the exit channel presumably closes to prevent backsliding of the substrate, while the titratable residues of the TMD remain protonated—the O* state is formed. The closing of the exit channel is likely facilitated by the neighbouring monomer adopting the O state, since a computational study suggests that the presence of two neighbouring O states results in a steric overlap in the PD[14] that likely occurs between the PN1 subdomain of one monomer and PN2 subdomain of its neighbour. Alternatively, the exit tunnel might spontaneously collapse after the substrate has left the channel. From the O* state the proton is released on the cytoplasmic side of the membrane, and deprotonation of the titratable residues in the TMD triggers the structural changes that lead to the L state as previously described[9,10,14]. The O* state might represent a local energy minimum in the OqxB structure which leads to the formation of O*O*O* in the absence of a substrate. The interactions of the RND pump with the periplasmic adaptor in the full tripartite assembly have been suggested to facilitate the transition of the O* to O state in MexB as part of the MexAB-OprM complex[23]. As structural data of OqxB in the tripartite complex are still missing, it cannot be excluded that this hypothesis might also apply for OqxAB-TolC.

## Discussion

Members of the OqxB cluster share a similar resistance phenotype[21,22,28–30] and their substrate preferences could be partially recreated in MdtF[27] and AcrB (this study) by a single V to F substitution in the DBP. Some of the effects of this substitution on the architecture of the DBP are likely shared between members of the OqxB cluster and the variants. These variants possess an additional aromatic residue for π-π-interactions, which could potentially enhance binding for certain substrates. Additionally, the reduced size of the DBP hinders the deeper penetration of the substrate into the pocket. For drugs exhibiting reduced resistance upon substitution, both these properties in the V612F/W variants might explain the decrease of pump activity for those drugs (Figs. 1c, 2 and Supplementary Figs. 5, 6). Further, a comparison of the porter domain of AcrB and OqxB shows that in the T states a shift of the PC2 subdomain towards the PC1 subdomain is observed in OqxB (Supplementary Fig. 22). As these subdomains flank the AP, this results in a smaller AP cleft in OqxB compared to AcrB. Additionally, the channels leading from the TMD to the DBP in OqxB have smaller bottleneck radii and are overall narrower compared to AcrB. A constriction of the channel connecting the AP and the DBP is also observed (Fig. 4c). Finally, in sharp contrast to AcrB that adopts different conformations with at least one L state monomer in the apo state, OqxB adopts the closed O* state. Currently, there is no experimental structure of OqxB in the L state, and it is unclear whether the protein can adopt this conformation. The generally narrower binding pockets and entrance channels of OqxB potentially limit the binding of high-molecular weight drugs, such as erythromycin, and thus evoke the substrate preference towards smaller and more flexible drugs, such as the phenicols, fluoroquinolones and linezolid. High-molecular-weight drugs are found associated with the L state in AcrB and initial binding to this state might be an important prerequisite for their entry in the PD interior[41]. Thus, some of the phenotype similarities between members of the OqxB cluster and the V612 variants of AcrB, like the reduced resistance against erythromycin, might be induced by a common effect of reduced initial binding of high-molecular weight drugs. In OqxB this is evoked by the narrow binding pockets and entrance channels, and potentially by the absence of a L state, whereas in the AcrB variants it is induced by the decrease of the fraction of monomers in the L state.

Regardless of whether OqxB and other cluster members might cycle through the T and O (and O*) states only, we anticipate that the coupling energetics will remain identical to AcrB and those cluster members. Both the L and T states in AcrB contain deprotonated D407 and D408 in the TMD, while in the O state these carboxylates are in a protonated state[14] (or at least one of them[42]). Therefore, protonation and deprotonation steps will be the same irrespective of whether the trimer cycles through the two states T and O and back to T (OqxB) or through the three states L, T, and O and back to L (AcrB), since there are no protonation/deprotonation events between the transition between L and T.

The data presented here for AcrB and OqxB, as well as previously published structural data[17,19–22,43] reveal a striking diversity in the conformations adopted by RND multidrug efflux pumps (Fig. 5 and Supplementary Fig. 23). On one side of the spectrum, AcrB adopts multiple trimer conformations in the apo state with an abundance of L monomers (Fig. 5, top panel). The equilibrium between those states can be modulated by interactions with substrates or by amino acid substitutions, as seen for the DDM solubilised AcrB samples and for the V612F/W variants, respectively. Binding of a substrate to the already open AP in the L state and DBP in the T state likely changes the existing equilibrium between these conformations and thus favours the formation of the LTO state. This mechanism is commonly referred to as conformational selection. On the other side of the spectrum, OqxB adopts a single trimer conformation, the O*O*O* state, in which all binding pockets and entrance channels are closed (Fig. 5, bottom panel). Substrates might interact with the entrance cleft of the AP, inducing the opening of the AP, or substrates might enter the PD from grooves in the TMD inducing the opening of the entrance channels and binding pockets in the PD interior. In both cases, the interactions with the substrate would induce the conformational changes required for the formation of the next state(s) of the conformational cycle, similarly to the mechanism referred to as induced fit. Thus, we provide here experimental evidence for the conformational selection mechanism of AcrB and show that an AcrB homologue from a different phylogenetic cluster, OqxB, might have a distinct mechanism of substrate binding.

The RND efflux pumps CmeB from *C. jejuni* and AdeB from *A. baumannii*, that bridge the AcrB and OqxB clusters show less conformational heterogeneity than AcrB and adopt the OOO state that is similar to the O*O*O* state of OqxB[18–20,43]. Nevertheless, they still adopt asymmetric conformations with monomers containing open substrate binding pockets in the apo state (Supplementary Fig. 23). We propose that the sequence features underlying the apo state configuration and thus the mechanism of substrate binding might be conserved in phylogenetic clusters and shared between close RND homologues. As demonstrated in the current work, changes in the conformational landscape contribute to changes in substrate specificity. Thus, the observed differences between the conformational landscape of RND multidrug efflux pumps might be one of the determinants of their substrate specificity spectrum.

## Methods

### Phylogenetic analysis of RND genes

For analysis of the conservation of deep binding pocket residues in a panel of Gram-negative bacteria, the representative protein sequences of the HAE-1 RND transporter family in the transporter classification database[44] (accessed 18.08.2023) with addition of the BpeF and CmeB sequences were analysed (Supplementary Table 1). A phylogenetic tree was created after a multiple sequence alignment with ClustalOmega[45] and visualised with iTOL[46]. Logo representations of the consensus sequence of the phylogenetic clusters were created with WebLogo[47]. Additionally, the same set of sequences was analysed by cc-analysis after a pairwise sequence alignment with PaSiMap[48].

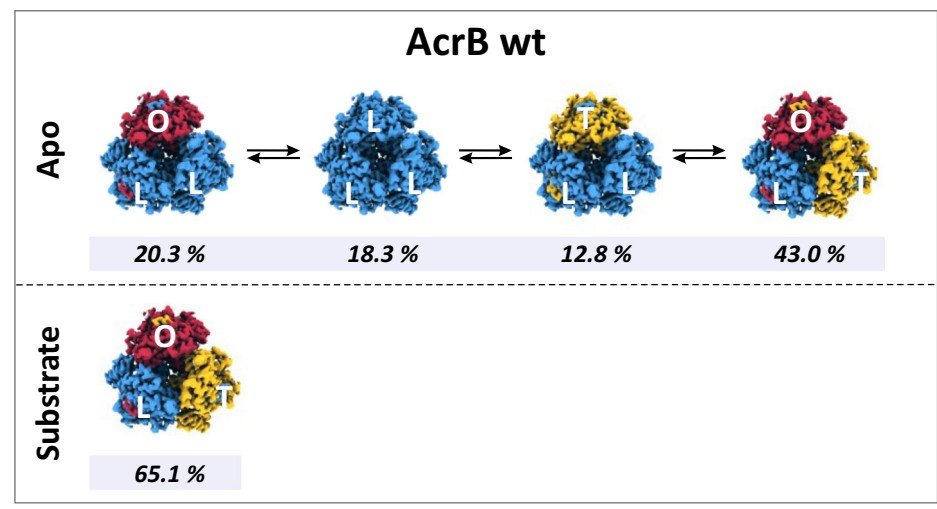

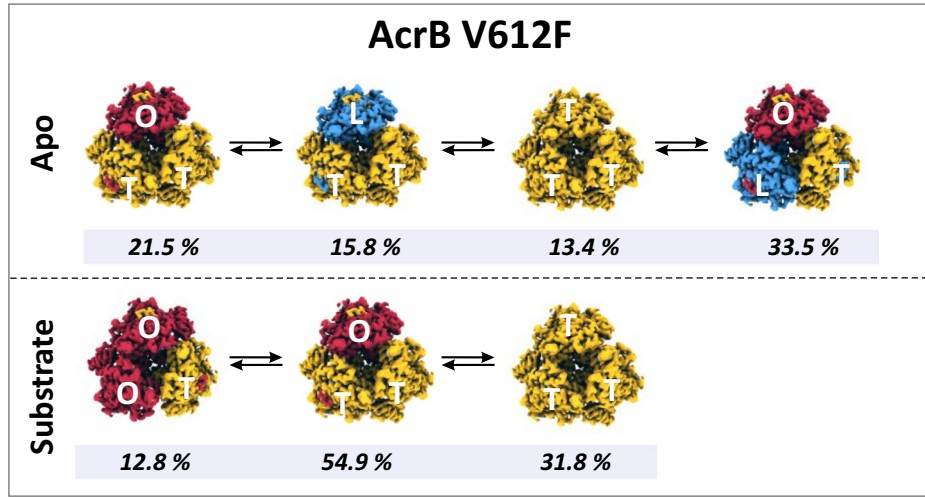

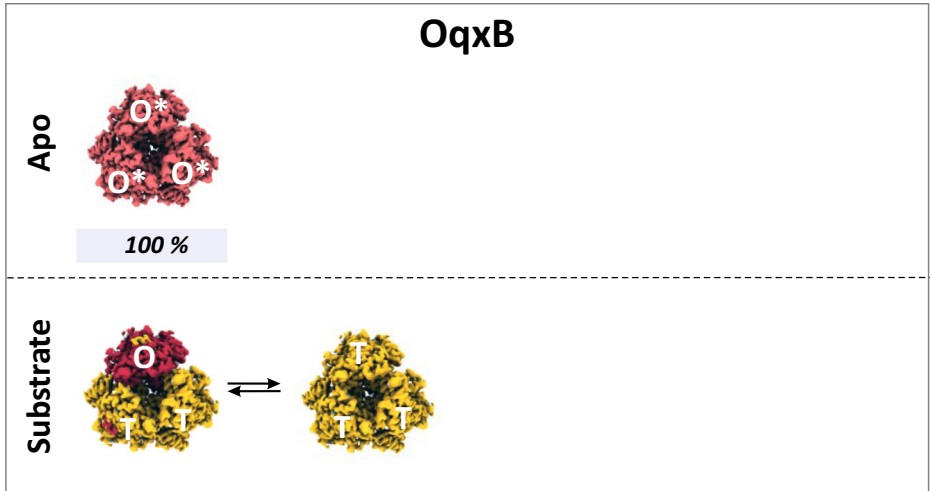

**Fig. 5 | Conformational states diversity in AcrB and OqxB.** In the absence of substrates (Apo-state), AcrB wildtype (wt) adopts several different conformations in equilibrium (top panel). Binding of the substrate in the already existing binding sites (AP in the L state and DBP in the T state) changes this equilibrium towards an increased occurrence of the LTO state (up to 65% of all particles, the occurrence of the other conformational states found in the apo-state were LLO, LLT and LLL was reduced to 8.6%, 3.4%, or 7.4%, respectively, see Fig. 3). The V612F substitution (AcrB V612F) on the other hand changes the equilibrium between the L and T states in the absence of substrate (Apo-state), thus inducing an increased frequence of T state containing trimers. In presence of substrate, the AcrB V612F trimers only adopt conformations where T and O conformations were detectable, the L state is absent. The trimers adopted the OTT, TTT, and OOT conformations, with an occurrence of 54.9%, 31.8%, and 12.8%, respectively (OOO was found for 0.5% of the particles, see Fig. 3). The latter conformational state distribution resembles the conformational states found for OqxB in the presence of substrate. Whereas OqxB adopts a single conformation, O*O*O*, in the absence of substrate (Apo-state), crystal structures indicated that OqxB adopts the OTT and TTT states in the presence of substrate (DDM). The interactions with a substrate likely induce the conformational changes required for the formation of the further states of the conformational cycle. The frequencies indicated for each state are derived from the experimental data in Fig. 3.

## Plasmids and sequences

*E. coli* AcrB and *K. pneumoniae* OqxB with C-terminal 6x-His-tag were expressed from the pET24 vector. AcrB-specific DARPin, clone 1108_19, with an N-terminal 6x-His-Tag, and saposinA with a N-terminal 6x-His-tag followed by a TEV cleavage site were expressed from the pQE and pNIC28-Bsa4 vectors, respectively. All constructs have been described previously[22,37,38,49].

## Bacterial strains and growth media

Phenotype characterisation was performed with an *E. coli* BW25113 *ΔacrB* strain. For expression of AcrB and OqxB, *E. coli* C43 (DE3) *ΔacrB* cells were used. For expression of DARPin *E. coli* XL1 Blue and for expression of saposinA *E. coli* Rosetta gami-2 (DE3) cells were used. For vector amplification and cloning purposes *E. coli* Mach1T1 cells were used. Cells were grown on LB agar plates (10 g/L tryptone, 5 g/L yeast extract, 10 g/L NaCl, 1.5% agar) or in liquid cultures in LB (10 g/L tryptone, 5 g/L yeast extract, 10 g/L NaCl) or TB (12 g/L tryptone, 24 g/L yeast extract, 2.31 g/l $KH_2PO_4$, 12.5 g/l $K_2HPO_4$, 0.4% (v/v) glycerol) medium containing an appropriate selection antibiotic (50 μg/mL kanamycin for pET24, 50 μg/mL carbenicillin for pQE, 50 μg/mL kanamycin and 34 μg/mL chloramphenicol for pNIC28-Bsa4).

## Plate dilution assays (PDA)

Chemically competent *E. coli* BW25113 *ΔacrB* cells were transformed with AcrB variants and cultured overnight at 37 °C on LB agar plates supplemented with 50 μg/mL kanamycin. Pre-cultures in LB medium with 50 μg/mL kanamycin were inoculated with a single clone and incubated overnight at 37 °C. A serial dilution of the overnight culture starting from $OD_{600}$ 1 to $OD_{600}$ $10^{-5}$ in 10-fold steps was prepared. The dilution series were spotted on LB agar plates containing selection antibiotic (50 μg/mL kanamycin) and an appropriate amount of the substrate of interest (Supplementary Table 2). Plates were incubated at 37 °C for 18 h before imaging. The assay was performed with at least three biological replicates. For each experiment a control plate without a substrate was prepared to ensure that all variants show equal growth in the absence of the substrate. The expression levels of all variants were validated by Western blot. For quantification of the results, the last dilution step for which growth was visible was averaged for all replicates and normalised to the wildtype (variant−wildtype).

## Minimal inhibitory concentration (MIC) determination

Overnight cultures of *E. coli* BW25113 *ΔacrB* cells transformed with AcrB variants were prepared as described for the PDA and diluted to $OD_{600}$ of 0.018. A serial dilution of the substrate of interest in twofold dilution steps was prepared in LB medium with 50 μg/mL kanamycin in a 96-well plate. 50 μL of the cell suspension was added to 100 μL of the serial dilution. The plates were incubated for 18 h at 37 °C. The $OD_{600}$ absorption of the plate was determined at a plate reader before (background absorption) and after the incubation at 37 °C. Background corrected $OD_{600}$ values higher than 0.18 were defined as growth and the MIC values corresponded to the lowest concentration of the substrate for which no growth was detected after the 18 h incubation. The MIC determination was performed in at least biological triplicates. MIC values were averaged for all replicates and normalised to the wildtype ($MIC_{variant}/MIC_{wildtype}$) (Supplementary Table 3).

## Whole cell accumulation assay

Overnight cultures of *E. coli* BW25113 *ΔacrB* cells transformed with AcrB variants were prepared as described for the PDA. 50 mL LB medium with 50 μg/mL kanamycin were inoculated with 500 μL overnight culture and incubated at 37 °C until $OD_{600}$ values of 0.7–0.9 were reached. Cells were harvested by centrifugation at 4000 × g and 4 °C for 5 min and washed with potassium phosphate (KPi) buffer (50 mM potassium phosphate pH 7.5, 1 mM $MgSO_4$). Cells were resuspended in KPi buffer supplemented with 0.2% glucose and the $OD_{600}$ was adjusted to 2.135 μL cells were added to 15 μL berberine solution in a black 96-well plate. Berberine accumulation was monitored for 40 min by measurement of the fluorescence at the excitation and emission wavelengths of 365 nm and 540 nm, respectively. The experiment was performed in biological triplicates.

## Protein expression

The expression of all constructs followed a similar procedure. A single clone of freshly transformed cells was used for inoculation of a pre-culture in LB medium with an appropriate selection antibiotic. The pre-culture was incubated overnight at 37 °C. 1 L medium (LB medium with 1% glucose for DARPin expression, TB medium for all other constructs) with an appropriate selection antibiotic was inoculated with 10 mL pre-culture and incubated at 37 °C under continuous shaking until an $OD_{600}$ value of 0.5--0.8 was reached. Expression was then induced with 1 mM isopropyl-beta-D-thiogalactopyranosid (IPTG). For expression of DARPin the culture was incubated at 37 °C for 4 h. For all other constructs the expression culture was incubated at 20 °C for 20 h. Cells were then harvested at 17,600 × g and 4 °C for 20 min. The cell pellet from the expression culture was resuspended in 20 mM Tris-HCl, pH 8.0, 0.5 M NaCl, 2 mM $MgCl_2$, 10 mg/L DNAseI and 0.2 mM PMSF and lysed with a Stansted SPCH-EP-10 pressure cell homogenizer (Homogenizing Systems Ltd, UK) at 22 kPsi. Cell debris in the lysate were removed by centrifugation at 20,000 × g and 4 °C for 20 min

## Purification AcrB and OqxB

The cell lysate prepared as described above was centrifuged at 186,000 × g and 4 °C for 1 h. The membrane pellet was resuspended in 4 ml 20 mM Tris-HCl pH 8.0, 0.5 M NaCl per g wet membrane weight, frozen in liquid nitrogen and stored at −80 °C until purification. The membrane suspension was diluted with the equal volume of IMAC wash buffer (20 mM Tris pH 7.5, 150 mM NaCl, 10% (v/v) glycerol) and imidazole was added to a final concentration of 20 mM. n-Dodecyl-β-D-maltopyranoside (DDM) was added to the final concentration of 1% for solubilisation and the membrane suspension was incubated at 4 °C for 1 h. Insolubilised lipids were removed by centrifugation at 186,000 × g and 4 °C for 30 min and the detergent extract was incubated with Ni-NTA beads, pre-equilibrated with IMAC wash buffer, for 1 h at 4 °C. The beads were washed three times with 15 column volumes of IMAC wash buffer supplemented with 0.02% DDM and containing 20 mM, 80 mM and 110 mM imidazole (AcrB) or 20 mM, 60 mM and 80 mM imidazole (OqxB). The sample was eluted with 10 column volumes IMAC wash buffer with 220 mM imidazole and 0.02% DDM, concentrated with an Amicon 100 Ultra-15 concentrator (100 kDa cutoff) and loaded on a Superose 6 10/300 increase column for size exclusion chromatography (SEC) in 20 mM Tris, pH 7.5, 150 mM NaCl, 0.02 % DDM. All purification steps were performed at 4 °C.

## Purification DARPin

The cell lysate prepared as described above was centrifuged at 137,000 × g and 4 °C for 1 h to remove cell debris and insoluble material. The supernatant was loaded on gravity flow Ni-NTA column pre-equilibrated with wash buffer. The resin was washed with 30 column volumes each of wash buffer (50 mM Tris-HCl pH 7.5, 0.4 M NaCl) containing 0 mM and 20 mM imidazole, respectively. The sample was eluted with 10 column volumes wash buffer with 250 mM imidazole and concentrated with an Amicon 100 Ultra-15 concentrator (10 kDa cutoff). During the concentration, the buffer was exchanged to 50 mM Tris-HCl pH 7.5, 0.4 M NaCl. The purified DARPin was divided into aliquots, frozen in liquid nitrogen and stored at −80 °C until further usage.

## Purification saposinA

The cell lysate prepared as described above was incubated for 10 min at 85 °C and precipitates were removed by centrifugation at 20,000 × *g* and 4 °C for 30 min. The supernatant was loaded on gravity flow Ni-NTA column pre-equilibrated with buffer. The resin was washed with 10 column volumes each of wash buffer (20 mM HEPES, pH 7.5, 150 M NaCl) containing 0 mM and 20 mM imidazole, respectively. The sample was eluted with 6 column volumes wash buffer with 100 mM imidazole, concentrated with an Amicon 100 Ultra-15 concentrator (3 kDa cutoff) and loaded on a Superose 6 10/300 increase column for SEC in 20 mM HEPES pH 7.5, 150 mM NaCl. Purified saposinA was digested with in-house produced TEV protease overnight at 4 °C to remove the 6x-His tag, then the sample was re-applied on the Ni-NTA resin. The flow-though was collected, concentrated, frozen in liquid nitrogen and stored at −80 °C until further usage.

## Reconstitution of AcrB and OqxB in salipro nanodiscs (SP-ND)

*E coli* total lipids (Avanti polar lipids) were dissolved in chloroform, the solvent was evaporated at a rotational evaporator and the lipid film was dissolved in 50 mM HEPES pH 7.5, 150 mM NaCl (final concentration lipids: 10 mg/mL) by sonication in an ultrasonic bad. The lipid stock was frozen in liquid nitrogen and stored at −80 °C until further usage.

Purified, His-tag cleaved saposinA was mixed with the lipid stock in a molar ratio of saposinA:lipids of 1:10 and the volume of the sample was adjusted to 1 mL with 50 mM sodium acetate, pH 4.8, 150 mM NaCl. The sample was incubated for 20 min at 37 °C and precipitates were removed by centrifugation at 20,000 × *g* for 10 min. The buffer was exchanged to 20 mM Tris, pH 7.5, 150 mM NaCl using a Sephadex G-25 gravity flow desalting column. The thus formed SP-ND were added to purified, DDM-solubilised AcrB or OqxB in the molar ratio AcrB/OqxB:saposinA:lipids 1:10:100. The volume of the sample was adjusted with detergent-free buffer so that the final DDM concentration is 0.01%. The sample was dialysed against 500 mL detergent-free buffer (20 mM Tris, pH 7.5, 150 mM NaCl) overnight at 4 °C and, after buffer exchange against fresh buffer, for further 3 h at 4 °C. Samples were then concentrated with an Amicon 100 Ultra-15 concentrator (100 kDa cutoff) and loaded on a Superose 6 10/300 increase column for SEC in 20 mM Tris, pH 7.5, 150 mM NaCl. SEC fractions containing the SP-ND reconstituted AcrB/OqxB were collected and concentrated to 1.5–3 mg/mL for cryo-EM grids preparation.

## Crystallisation, X-ray data collection and analysis

For crystallisation of AcrB in the presence of DARPins, purified, DDM-solubilised AcrB and DARPins were mixed in the molar ratio of 1:2 and incubated for 20 min at 4 °C. Excess DARPin was removed by SEC in 20 mM Tris pH 7.5, 150 mM NaCl, 0.03 % DDM and samples were concentrated to 10–15 mg/mL. For co-crystallisation with minocycline, the substate was added to the final concentration of 2 mM. Crystals were grown by the hanging drop vapour diffusion method in 24-well plates with 1 mL reservoir solution for 1–2 weeks at 18 °C. Asymmetric V612N crystals (LTO state) were obtained from 50 mM N-(2-acetamido)iminodiacetic acid (ADA), pH 6.6, 5% (v/v) glycerol, 6–9% (w/v) polyethylene glycol (PEG) 4000, 110–220 mM (NH₄)₂SO₄. Symmetric V612N (TTT state) were obtained from 0.1 M MES pH 6.5, 5.5-20.5 % (v/v) PEG400. Apo V612W crystals were obtained from 0.1 M sodium acetate pH 4.5, 0.1 M NaCl, 0.1 M MgCl₂, 20–37.5% (v/v) PEG400. V612W crystals with minocycline were obtained from 0.1 M MES pH 6.5, 5.5–20.5% (v/v) PEG400. V612F crystals with minocycline were obtained from 0.1 M sodium acetate pH 4.5, 3–7% (v/v) PEG200, 15–25 % (v/v) PEG400, 0.15 M MgCl₂, 0.15 M NaCl. For crystallisation of apo V612F in the absence of DARPins, purification and crystallisation were carried out with cyclohexyl-n-hexyl-β-D-maltoside as detergent as previously described[10]. Clarithromycin was added to the sample with a final concentration of 1.2 mM prior to crystallisation,

but no ligand densities were observed in the structure, thus resulting in an apo structure of AcrB. Crystals were obtained from 0.1 M citrate pH 4.6, 5% (v/v) PEG400, 16–21% (v/v) PEG300, 8–11 % (v/v) glycerol. Crystals from the ADA and citrate screens were cryo-protected with 28% (v/v) glycerol, all other crystals were cryo-protected in 20–30% (v/v) PEG400. Purified OqxB was prepared as described previously[22]. OqxB crystals were grown by the sitting drop vapour diffusion technique at 25 °C. Protein solution was mixed (1:1) with reservoir solution containing 12% PEG4000, 0.2 M MgCl₂, 100 mM ADA (pH 6.5). Crystals were grown within 1–2 weeks to optimal size (0.3 × 0.3 × 0.5 mm³). The concentration of glycerol was gradually increased to 30% (v/v) by soaking in several steps for optimal cryo-protection. Crystals were picked up using nylon loops (Hampton Research, CA, USA) for flash-cooling in cold nitrogen gas from a cryostat (Rigaku, Japan).

X-ray diffraction data of AcrB crystals were collected at the beamlines X06SA and X10SA of the Swiss Light Source (Paul Scherrer Institut, Villigen, Switzerland) and P13 of the Deutsches Elektronen Synchrotron (Hamburg, Germany). OqxB data sets were collected at 100 K using an EIGER hybrid photon-counting (HPC) pixel-array detector (Dectris, CH) on the BL44XU beamline at SPring-8 (Sayo, Japan).

Diffraction data were processed with XDS (BUILT = 20220110)[50] and the programmes from the Phenix package v1.20.1[51,52]. The crystal structures were solved by the molecular replacement method using MOLREP v11.0[53] and Phaser v2.7[54]. The AcrB (PDB ID: 4dx5) and OqxB (PDB ID: 7cz9) structures were used as the search models. Automated structure refinement was performed with Refmac v5[55] and phenix.refine v1.20.1[56]. Model building was performed with Coot v0.9[57]. MolProbity v4.02[58] was used for structure validation. Data collection and refinement statistics are summarised in Supplementary Tables 6–9. Figures were generated with ChimeraX v1.6[59].

## Cryogenic electron microscopy (cryo-EM) sample preparation, data collection and analysis

All cryo-EM samples were applied on glow-discharged R1.2/1.3, 300-mesh Cu holey carbon grids (Quantifoil Micro Tools GmbH) and plunge-frozen in liquid ethane using a Vitrobot Mark IV (Thermo Scientific, Waltham, USA). Samples were vitrified at 100 % humidity and 4 °C after blotting with Whatman papers (grade 595) that were pre-equilibrated in the Vitrobot for 1 h. DDM-solubilised samples were vitrified with nominal blotting force of -25, blotting time of 6-10 s and waiting time of 40 s. SP-ND samples were vitrified with blotting force of -3, blotting time of 4-8 s and waiting time of 40 s.

DDM-solubilised AcrB wildtype (1.5 mg/mL) and V612F (1.8 mg/mL) samples were recorded on a FEI Titan Krios cryo-TEM (Thermo Scientific, Waltham, USA) operating at 300 kV in nanoprobe EFTEM equipped with a K2 summit direct detector (Gatan Inc., Pleasanton, USA) and a post-column energy filter (GIF Quantum SE, Gatan) operating in zero-loss mode with a slit width of 20 eV. Data were recorded using Serial-EM[60] at ×105000 magnification (1.05 Å pixel size) with defocus values of −0.8 to −3.5 μm. Dose-fractionated movies were acquired in counting mode with a dose rate of 8 e⁻/Å²s⁻¹ and 50 e⁻/Å² total dose per micrograph.

The SP-ND V612F (2.8 μg/mL) dataset was acquired on a Titan Krios cryo-TEM (Thermo Scientific, Waltham, USA) operating at 300 kV equipped with a BioQuantum-K3 imaging filter (Gatan Inc., Pleasanton, USA) and a post-column energy filter (GIF Quantum SE, Gatan) operating in zero-loss mode with a slit width of 20 eV. Data were recorded using Serial-EM v3.8[60] at ×130000 magnification (0.68 Å pixel size) with defocus values of −0.5 to −3.0 μm. Dose-fractionated movies were acquired in counting mode with a dose rate of 16 e⁻/Å²s⁻¹ and 60 e⁻/Å² total dose per micrograph.

DDM-solubilised AcrB V612W (1.9 μg/mL), SP-ND AcrB wildtype (2.5 μg/mL) and OqxB (2.7 μg/mL) datasets were acquired on Titan

Krios G3i (Thermo Scientific, Waltham, USA) operating at 300 kV, equipped with a BioQuantum-K3 imaging filter (Gatan Inc., Pleasanton, USA) operated in EFTEM mode with a zero-loss peak slit width of 30 eV. Data were recorded using EPU v2.12 (Thermo Scientific, Waltham, USA) with nominal magnification 105000x (0.837 Å pixel size) and defocus values of −0.8 to −3.5 μm (V612W) and −0.8 to −2.4 (AcrB wildtype and OqxB). Data were acquired as dose-fractionated movies with 50 e$^-$/Å$^2$s$^{-1}$ total dose per image, equally distributed over 50 fractions.

Cryo-EM data analysis was performed with cryoSPARC v3.2[61] and Relion v4.0[62]. The general processing pipeline is depicted in Supplementary Fig. 7 and the processing of each individual dataset in explained in more details in Supplementary Figs. 8–15. In brief, first beam-induced motion correction and CTF estimation were performed. For initial particle picking a blob picker with particle diameter of 100-160 Å was used in cryoSPARC. After ab initio reconstitution a 3D reference was created and used for template-based automated particle picking. In Relion, either approximately 1000 particles were picked manually and used to create a 2D reference for template-based picking; or a 3D reference of one of the already processed datasets was directly used for template-based picking. Several iterative rounds of 2D classification were performed to remove false-positive picks and poor-quality particles. After 3D map reconstruction, a 3D classification was performed to further cure the dataset of poor-quality particles. CTF refinement, local correction of the beam-induced motion and 3D refinement of the trimeric particles without imposed symmetry were performed. Monomers were extracted from the trimers in Relion utilising the C3 pseudosymmetry through the central axis of AcrB and OqxB as described previously[19]. The 3D volumes were processed with C3 symmetry and a C3 symmetry expansion was performed. This triplicates the particles and rotates them along the symmetry axis so that all three monomers of each trimer are aligned at the same position. A soft monomer mask created based on the AcrB (PDB ID: 4dx5) or OqxB (PDB ID: 7cz9) models was used to subtract two of the monomers. The resulting monomer volume was subjected to several rounds of 3D classification with a varying number of classes (minimal 3). The goal was to obtain the maximum number of classes with the best resolution. The classification utilised the monomer mask used for the subtraction and a low pass filtered trimer volume as the reference map and was performed without image alignment and with a regularisation parameter T of 15. The 3D classes were refined, and the conformational state of each class was determined by comparison with each monomer (L, T and O) of the asymmetric AcrB structure (PDB ID: 4dx5), based on characteristic structural features like the position of the subdomains in the porter domain. A custom MATLAB v9.13 (The MathWorks Inc., Natick, Massachusetts, USA) script was used to calculate the trimer composition of the sample based on the position of the extracted monomers.

Structure models of the best resolved O state monomers of AcrB were build based on the experimental structure of AcrB in the O state (PDB ID: 4dx5). The structure model of OqxB was based on the AlphaFold[63] predicted structure available under Uniprot accession number U5U6L7 (accession date: January 8th, 2024, Supplementary Fig. 24). Structure refinement was performed with phenix.real_space_refine v1.20.1[56], Coot v0.9[57] and ISOLDE v1.7[64]. MolProbity v4.02[58] was used for structure validation. Images of portions of the electron density maps and omit maps for minocycline-bound AcrB V612F and V612W are shown in Supplementary Figs. 25 and 26. Data collection and refinement statistics are summarised in the Supplementary Table 10. Figures were generated with ChimeraX v1.6[59].

## Reporting summary
Further information on research design is available in the Nature Portfolio Reporting Summary linked to this article.

## Data availability
The crystallographic structures generated in this study have been deposited in the PDB database under the following accession codes: AcrB V612W with bound minocycline: 9FE2, AcrB V612W apo: 9FE3, AcrB V612F with bound minocycline: 9FHC, AcrB V612F, apo: 9FE4, AcrB V612N (TTT state): 9FHJ, AcrB V612N (LTO state): 9FHG, OqxB (TTO state): 8ZXS. The cryo-EM structures generated in this study have been deposited in the PDB and EMDB database under the following accession codes: OqxB in salipro nanodiscs: 9FDZ, EMD-50334; OqxB monomer classes: EMD-50335, AcrB V612F monomer in the O state: 9FDQ, EMD-50332; AcrB V612W monomer in the O state: 9FDP, EMD-50331; AcrB wildtype in DDM: EMD-50328; AcrB V612F in DDM: EMD-50329; AcrB wildtype in salipro nanodiscs: EMD-50645. Single particle cryo-EM maps (.mrc format) as shown in the Supplementary Figs. are available for download under https://doi.org/10.6084/m9.figshare.28255283. For reference and comparison, the previously published structures available in the PDB database under following accession codes have been used: 4dx5, 1iwg, 7cz9, 5o66, 6zoe, 7wls, 7kgh, 6ta6, 6t7s, 5lq3. For reference for model building the predicted structure available in the AlphaFold Structure Database under UniProt accession number U5U6L7 was used. The PDB files of the top docking poses with chloramphenicol, minocycline, doxorubicin, and erythromycin on the *E. coli* efflux pump AcrB wildtype, V612F and V612W variants have been deposited in the zenodo database under following accession code: https://doi.org/10.5281/zenodo.14719970. Molecular dynamics simulation trajectories of chloramphenicol and minocycline in complex with wildtype AcrB and the V612F and V612W variants have been deposited in the zenodo database under following accession code https://doi.org/10.5281/zenodo.15367315. Source data and raw data for the plate dilution assay, MIC determination and the accumulation assay are available as additional source and raw data files. Unless otherwise stated, all data supporting the results of this study can be found in the article, supplementary, and source data files. Source data are provided with this paper.

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

## Acknowledgements

We thank Dr. Anja Seybert (Buchmann Institute for Molecular Life Sciences and Institute for Biophysics, Goethe University Frankfurt, Germany) as well as the Central Electron Microscopy Facility (Max-Planck-Institute of Biophysics, Frankfurt, Germany), in particular Dr. Sonja Welsch and Dr. Simone Prinz, for the technical and scientific support during cryo-EM sample preparation and data acquisition. We thank Dr. Fabrizio C. Muredda and Andrea Bosin (University of Cagliari, Italy) for technical support in setting up local computational facilities. K.M.P. acknowledges support by DFG-SFB807, DFG-SFB1507, DFG-EXEC-115, and Pfizer ASPIRE grant. A.S.F. acknowledges support by DFG-EXEC-115 and DFG FR 1653/14-1. S.M., U.O. and E.Y. acknowledge support by JSPS KAKENHI Grant Numbers JP21H02412 (S.M.), JP22K06099 (U.O.) and JP22H02558 (E.Y.). This research was partially supported by the Platform Project for Supporting Drug Discovery and Life Science Research, Basis for Supporting Innovative Drug Discovery and Life Science Research (BINDS) from AMED (JP20am0101072) and the Joint Research Committee of the Institute for Protein Research, Osaka University. Synchrotron radiation experiments were performed at BL44XU of SPring-8 (2019A6500, 2019A6700, 2019B6500, 2019B6700). M.A. and A.V.V. gratefully acknowledge the "One Health Basic and Translational Research Actions addressing Unmet Needs on Emerging Infectious Diseases (INF-ACT)" foundation by the Italian Ministry of University and Research, PNRR, mission 4, component 2, investment 1.3, project number PE00000007 (University of Cagliari). AVV acknowledges funding from the National Recovery and Resilience Plan (NRRP), Mission 4 Component 2 Investment 1.5—Call for tender No.3277 published on December 30, 2021, by the Italian Ministry of University and Research (MUR) funded by the European Union—NextGenerationEU. Project Code ECS0000038—Project Title eINS Ecosystem of Innovation for Next Generation Sardinia—CUP J85B17000360007—Concession Decree No. 1056 adopted on June 23, 2022, by the Italian Ministry of University and Research (MUR). M.A. and A.V.V. received financial support by the NIAID/NIH grant no. R01AI136799.

## Author contributions

M.L. performed and analysed the sequence similarity comparison of RND efflux pumps. T.E. and H.C. established the phenotype screening pipeline for the AcrB mutants. M.L. and I.M.S. performed the plate dilution assays and determined the minimal inhibitory concentration for all substrates for all technical and biological repeats. M.L. performed the whole cell transport assay. ML analysed all phenotype data. M.L. and T.E. expressed and purified the AcrB V612F (T.E.), and V612W and V612N (M.L.) samples for crystallisation and performed the crystallisation experiments. M.L., T.E., and K.D. acquired X-ray diffraction data and built and refined the respective structural models. U.O. established the OqxB overexpression system. S.M. purified OqxB for crystallisation and performed the crystallisation experiment and E.Y. performed the X-ray diffraction experiment. S.M. performed the crystallographic analysis of the OqxB_TTO structure. M.L. and H.Z. expressed and purified AcrB wildtype (H.Z.), and V612F and V612W (M.L.) samples in DDM and prepared the grids for cryo-EM analysis. M.L., H.C., and C.B. acquired the cryo-EM datasets and M.L. and C.B. analysed the data. M.L. built and refined the O-state models of V612F and V612W. M.B. established the reconstitution protocol for the samples solubilised in salipro nanodiscs. M.L. and M.B. expressed and purified the AcrB wildtype (M.L.) and OqxB (M.L. and M.B.) samples in nanodiscs and prepared the cryo-EM grids. M.L. acquired the cryo-EM datasets, analysed the data and built and refined the OqxB structural model. M.L., M.A., and A.V.V. prepared and performed the docking and free binding energy calculation and analysed the results. M.A. and A.V.V. conducted the molecular dynamics simulations and analysed the results. M.L., T.E., K.M.P., A.V.V., K.D., and A.S.F. were involved in the conception and design of the experiments, and the analysis and interpretation of the data. K.M.P., A.S.F., S.M., U.O., and E.Y. are holders of the grants funding the experiments. M.L. and K.M.P. wrote the manuscript. All authors edited the manuscript.

## Funding

## Competing interests

The authors declare no competing interests.
