## [Transparent Peer Review file · Nature Communications]

Conformational plasticity across phylogenetic clusters of RND multidrug efflux pumps and its impact on substrate specificity

Corresponding Author: Professor Klaas Pos

Version 0:

Reviewer comments:

Reviewer #1

(Remarks to the Author)

This manuscript presents a well-executed and insightful study on the impact of deep binding pocket (DBP) mutations (V612F/W) in AcrB and their effects on substrate specificity and antibiotic resistance. The combination of structural biology, computational chemistry, and functional assays provides a comprehensive mechanistic view of how these mutations alter the pump's conformational landscape and drug efflux efficiency.

The study's strengths lie in its rigorous structural and functional characterization, clearly demonstrating how V612 mutations alter AcrB's conformational equilibrium. Cryo-EM and X-ray crystallography provide direct evidence of these structural changes, while the comparison with OqxB from *K. pneumoniae* adds valuable evolutionary context. Antibiotic resistance profiling through MIC assays establishes a clear link between structural modifications and resistance phenotypes, further supported by berberine accumulation assays confirming reduced efflux efficiency. Computational modeling, including ensemble docking and MM/GBSA calculations, aligns well with experimental data, reinforcing observed substrate binding trends.

MAIN TEXT

Suggestions/Questions/Areas of Improvement

1. Unclear Functional Implications of the OOO State in OqxB

The cryogenic electron microscopy (cryo-EM) data show that OqxB is locked in an OOO state, which differs from AcrB's usual LTO cycling.

Would an OqxB conformational equilibrium be altered in the presence of substrates?

Consider discussing whether the OOO state is truly physiological or if additional structural studies in substrate-bound conditions could provide more clarity.

2. The Role of Protonation States in Transport Function

AcrB is a proton-coupled antiporter, and its function is highly dependent on protonation dynamics of residues like E130, D407, and D408. Did the authors explore whether V612F/W mutations alter the proton transport cycle?

SUGGESTION: Given that proton-driven conformational cycling is essential for substrate extrusion, some discussion of how these mutations might affect protonation equilibria would be helpful.

3. The Functional Impact of TTT State Accumulation

The study suggests that V612F/W mutations push AcrB into a TTT-dominated state, likely reducing efflux efficiency.

CLARIFY & INCLUDE: Does this mean that transport is simply slower, or could it indicate a functional impairment where

drugs get trapped in the DBP without extrusion?

Computational Studies

The computational section in the supplementary file is well-structured and provides valuable insights into how V612 mutations in AcrB affect drug binding. The use of homology modeling, ensemble docking, and MM/GBSA free energy calculations strengthens the study and complements the experimental findings. However, some areas could benefit from additional clarification or validation.

Suggestions/Questions/Areas of Improvement

1. Lack of Molecular Dynamics (MD) Simulations

While ensemble docking captures some flexibility, it does not account for explicit solvent effects or long-range protein-ligand interactions. The authors reference previous MD studies on chloramphenicol flipping in the DBP, but an explicit MD analysis of the V612 mutants could provide dynamic insights into ligand retention and transport efficiency.

CLARIFY AND INCLUDE: A short (100–200 ns) MD simulation of wildtype and V612 mutants with minocycline, doxorubicin, and chloramphenicol could confirm ligand stability and adaptive binding site changes.

2. Clarification Request on Pairwise RMSD Calculations and Symmetry Corrections

In the homology modeling and docking ensemble preparation, pairwise RMSD calculations were used to select structures that capture significant conformational diversity (RMSD > 1 Å). However, AcrB is a homotrimeric protein, and symmetry-related residues across different monomers may contribute to these RMSD values.

CLARIFY: Did the authors apply symmetry corrections to ensure equivalent residues in different protomers are properly aligned before computing RMSD?

INCLUDE: If not, would applying a symmetry-corrected alignment (e.g. ProDy, TM-align, or MDAnalysis symmetry-aware RMSD tools) affect the structural selection process?

3. No Alchemical Free Energy Perturbation (FEP) Calculations

CLARIFY: Have the authors considered running FEP calculations? MM/GBSA provides an approximate binding free energy, but it does not explicitly model ligand-water exchange, which is critical for efflux pumps. Since V612 mutations affect water accessibility in the DBP, FEP calculations would provide a rigorous $\Delta\Delta G$ mutation effect. There are now a lot of open source softwares that do that, consulting <https://openfree.energy/> maybe a good starting point.

4. Docking Poses for Erythromycin Might Be Overestimated

The MM/GBSA results suggest that erythromycin binds similarly in wildtype and mutants, yet experimental resistance data suggests altered efflux efficiency.

Could erythromycin binding be an artifact of docking?

5. Missing Water Network Analysis in the Deep Binding Pocket

Since the DBP undergoes conformational shifts, a solvent network analysis could provide insights into water displacement effects in V612 mutants.

FOR FUTURE: MD-based water occupancy maps (using GIST or WaterMap) could identify key hydration sites. This would help explain differences in drug retention in wildtype vs. mutant AcrB.

Reviewer #2

(Remarks to the Author)

This manuscript presents a comprehensive and detailed study on the conformational plasticity of RND multidrug efflux pumps and its impact on substrate specificity. The authors have conducted a series of well-designed experiments, including sequence analysis, structural determination, and functional assays, to elucidate the role of conserved residues in the deep binding pocket (DBP) of RND proteins. The findings are novel and provide valuable insights into the mechanisms of multidrug resistance in Gram-negative bacteria. The study is well-executed, and the data are robust and convincing. The manuscript is well-written and organized, making it accessible to a broad audience in the field of microbiology and structural biology. I recommend this manuscript for publication in its current form, with minor revisions to enhance clarity and context.

Minor Suggestions for Improvement:

1. Clarity of the Introduction:

While the introduction provides a good overview of the background and significance of the study, it could benefit from a more concise summary of the specific objectives and hypotheses being tested. This would help readers quickly grasp the main focus of the research.

2. Additional Context for Computational Studies:

The computational docking and free binding energy calculations are briefly mentioned in the results section. It would be helpful to include more details about the methods and assumptions used in these calculations. This would allow readers to better evaluate the reliability and relevance of the computational results.

3. Data Interpretation:

The authors mention that the V612F/W substitutions alter the binding poses of chloramphenicol and doxorubicin. It would be beneficial to include a more detailed discussion of the potential implications of these changes for the overall substrate specificity and resistance profile.

4. Figure and Table Legends:

Some of the figure and table legends are quite detailed. It would be advisable to simplify them in order to improve clarity and conciseness.

Reviewer #3

(Remarks to the Author)
[See attached document.]

Reviewer #4

(Remarks to the Author)

Lazarova et al. identified two phylogenetic clusters of RND multidrug efflux pump and demonstrated a single substitution of V612 residue can switch the resistance phenotype with multiple Cryo EM and crystallographic structures together with physiological data.

The information would be helpful insight to understanding the diverse evolution fundamental and clinical mutation specificity of the RND efflux pump.

The manuscript is well written and technique sound.

The author presented the entry and exit channels in AcrB and OqxB structures in Figure 4, are those channel radius correlated to the size of the drugs? Since the substrates can enter the PD through several channels, are there any evidences from the intermedia state of the Cryo EM structure.

Version 1:

Reviewer comments:

Reviewer #1

(Remarks to the Author)

I appreciate the detailed rebuttal and careful revisions throughout the manuscript, supplementary materials, and MD simulation checklist. Overall, I find that my earlier comments have been addressed satisfactorily. The manuscript is now substantially clearer, more rigorous, and presents a convincing mechanistic interpretation. Below I summarize my assessment:

1. OqxB OOO state and mechanistic implications

You have added a thoughtful comparison between AcrB and OqxB, including the new Figure 5, which clarifies the distinction between conformational selection (AcrB) and induced-fit-like opening (OqxB). While the physiological frequency of OOO cannot be fully established, you appropriately acknowledge this limitation. This addition satisfactorily resolves my concern.

2. Protonation states and coupling to the transport cycle

The added discussion explains the state-dependent protonation scheme and how it relates to the L/T/O cycle. The handling of the V612F/W variants is now well justified. The protonation choices in the MD simulations are also clearly described in the supplementary methods. This is a significant improvement.

3. Functional meaning of TTT accumulation

Your revision now connects the TTT bias observed for variants to impaired transport of substrates that require L-state entry (such as erythromycin). While you stop short of new kinetic measurements, you note this as an important future step. This framing is sufficient for the scope of the present work.

4. Substrate docking and erythromycin case

The docking and MM/GBSA analysis show that erythromycin binding poses are largely unaffected by V612F/W, while the global conformational redistribution explains the phenotype. This argument is now clearly articulated, and the supplementary results support your conclusion.

5. Computational rigor and transparency

The addition of three independent 250 ns replicas per system, with RMSD analyses and structural comparisons, considerably strengthens the computational results. The level of methodological detail is now adequate for reproducibility, and I appreciate that the data have been deposited publicly.

6. Editorial and figure clarifications

The figure references, arrows, and new comparative model figure address the clarity issues I raised previously. The manuscript now reads much more smoothly.

Minor suggestions for clarity (non-essential)

Consider adding a one-line explicit limitation statement in the main text noting that the physiological frequency of the OOO state is uncertain, even though the mechanism is plausible.

In the supplementary, a short justification for using the docked erythromycin pose as the RMSD reference (rather than an experimental CAM structure) would preempt potential confusion.

Since MM/GBSA is presented without entropic terms, a brief caveat in the main text (not just in the supplementary) that these values are qualitative/relative would be helpful.

Reviewer #2

(Remarks to the Author)

The authors have addressed all of my concerns. I have no further comments.

Reviewer #3

(Remarks to the Author)

This reviewer finds that all of the major and minor concerns have been adequately addressed in the revision. I am pleased to recommend its publication.

Reviewer #4

(Remarks to the Author)

The author has addressed all the comments.
